# Modeling Endometrium Biology and Disease

**DOI:** 10.3390/jpm12071048

**Published:** 2022-06-27

**Authors:** Nina Maenhoudt, Amber De Moor, Hugo Vankelecom

**Affiliations:** Unit of Stem Cell Research, Cluster of Stem Cell and Developmental Biology, Department of Development and Regeneration, Leuven Stem Cell Institute, 3000 Leuven, Belgium; nina.maenhoudt@kuleuven.be (N.M.); amber.demoor@student.kuleuven.be (A.D.M.)

**Keywords:** endometrium, stem cells, organoids, endometriosis

## Abstract

The endometrium, lining the uterine lumen, is highly essential for human reproduction. Its exceptional remodeling plasticity, including the transformation process to welcome and nest the embryo, is not well understood. Lack of representative and reliable study models allowing the molecular and cellular mechanisms underlying endometrium development and biology to be deciphered is an important hurdle to progress in the field. Recently, powerful organoid models have been developed that not only recapitulate endometrial biology such as the menstrual cycle, but also faithfully reproduce diseases of the endometrium such as endometriosis. Moreover, single-cell profiling endeavors of the endometrium in health and disease, and of derived organoids, start to provide deeper insight into cellular complexity and expression specificities, and in resulting tissue processes. This granular portrayal will not only help in understanding endometrium biology and disease, but also in pinning down the tissue’s stem cells, at present not yet conclusively defined. Here, we provide a general overview of endometrium development and biology, and the efforts of modeling both the healthy tissue, as well as its key diseased form of endometriosis. The future of modeling and deciphering this key tissue, hidden inside the womb, looks bright.

## 1. Introduction

Clearly, the uterus is an essential organ in human reproduction. Its inner lining, the endometrium, holds exceptional remodeling capacity, undergoing monthly cycles of growth (proliferative phase), differentiation (secretory phase), degeneration (menstrual phase) and regeneration with the restart of the cycle (Figure 1). Each phase displays specific morphological rearrangements and expression profiles in preparation for the endometrium’s prime function, that being welcoming and nesting the fertilized egg (blastocyst) so that pregnancy can successfully develop. The endometrium is built up of multiple cell types, including epithelial, stromal, immune and endothelial cells [1] (Figure 1). The menstrual cycle is meticulously regulated by the two primary ovarian steroid hormones, estrogen and progesterone, eventually culminating in an endometrium which is receptive to embryo implantation during the ‘window of implantation’ (WOI), established at the mid-secretory phase (6–10 days after ovulation) (Figure 1). In preparation of this phase, decidualization occurs to accommodate and nurture the implanting embryo. During this tissue transformation process, stromal cells differentiate into secretory (decidual) epithelioid cells that relay multiple molecular signals to prime the endometrium toward the receptive state, while glandular epithelial cells enhance their secretory activity, and epithelial cells facing the uterine lumen (luminal epithelium; Figure 1) alter their molecular and morphological features to enable the embryo to attach [2].

The human endometrium sits hidden inside the womb, which strongly hinders its detailed exploration. Insights in human endometrium biology and pathology have been mainly transposed from experimental animals and not highly fitting cell culture surrogates. Representative and reliable study models remain the most important hurdle to progress in understanding the human endometrium. Recently, organoid models have been established from both healthy and diseased endometrium [1,2,3]. Organoids represent in vitro cell constructs that mimic key characteristics of the originating tissue at both the phenotypical and functional level. These miniaturized organs develop from the tissue’s stem cells when exposed to the extracellular matrix (ECM) and (embryonic) signaling cues that play a role in the natural tissue in vivo. Organoids can be largely expanded and hold great promise for mechanistic exploration of both healthy and diseased tissues, for biobanking, drug screening and eventually also personalized medicine. Considering these important assets, organoids may substantially leverage our knowledge on endometrial development, functioning and disease.

Here, we briefly describe endometrium development and biology, thereby including recent single-cell (sc) profiling efforts, and give an overview of approaches used to model the endometrium and its key disease of endometriosis, together with potential applications of these models, including the pinpointing of endometrial stem cells, at present still not conclusively defined.

## 2. Endometrium Development and Biology

### 2.1. Endometrium Development

Knowledge on endometrium development and differentiation (maturation) mainly comes from experimental animals, in particular from mouse studies, which have helped in gaining insight into the genes and molecular pathways involved. The uterus is part of the female reproductive tract (FRT); the lower genital tract comprises the vagina and vulva, and the upper tract consists of the cervix, uterus, fallopian tubes and ovaries. The reproductive tract is part of the urogenital system, which also comprises the kidneys, gonads and urinary tract. During embryogenesis, the reproductive tract starts to develop after gastrulation through the differentiation of the intermediate mesoderm. Expansion and mesenchymal-to-epithelial transition (MET) generate the epithelial tubules, composed of two pairs of Wolffian (mesonephric) and Müllerian (paramesonephric) ducts (MD) (Figure 2A). After sexual differentiation, triggered by an interplay of hormones, the FRT develops from the MD (Figure 2B) while the Wolffian duct regresses, whereas the opposite occurs in males [4,5]. 

Multiple transcription factors (TFs) and signaling molecules are involved in MD formation as revealed by mouse studies (Table 1). Paired-box gene 2 (PAX) is essential in MD development, and knockout in female mice leads to absence of the FRT [6]. LIM1 (also known as LHX1) is expressed in the epithelium of the MD. In *Lim1* null female mice, the oviducts, uterus and upper portion of the vagina are missing [7]. Empty spiracles homeobox 2 (EMX2) is also essential, since *Emx2* knockout female mice lack the MD and hence also the reproductive tract, as well as the kidneys [8]. Retinoids, the active metabolites of vitamin A, play a crucial role during FRT development since female mice with compound mutations in retinoic-acid receptor (RAR) genes completely lack the FRT because of incorrect MD formation [9]. Similarly, discs large homolog 1 (*Dlgh1*) null mutants show defective fusion and impaired caudal elongation of the MD, leading to aplasia of the cervix and vagina [10]. Wingless-type MMTV integration site (WNT) signaling is also highly implicated in MD formation. WNT4 is required for early tubule formation of the MD, and *Wnt4* null female mice lack the FRT [11]. In addition, WNT9B is required for tubule development and caudal extension of the MD, and null mutants lack the uterus and upper vagina [12].

After formation and elongation, the MDs differentiate to form the oviducts (fallopian tubes), uterus, cervix and upper part of the vagina along the anterior–posterior axis. The proximal parts of the MDs remain separated and form the fallopian tubes, whereas the distal parts fuse into a single canal. The septum between the MDs resorbs, finally forming the uterus, cervix and upper vagina (Figure 2C,D). During these developmental stages, genes from the Homeobox (HOX) and WNT family are especially important. Along the anterior–posterior axis, a number of *Hox* genes are sequentially expressed, needed for the correct patterning of the FRT. *Hoxa9* is found in the oviduct, *Hoxa10* in the uterus, *Hoxa11* in the uterus and cervix, and *Hoxa13* in the cervix and upper vagina [42]. Maintenance of *Hoxa10* and *Hoxa11* expression during FRT development is regulated through WNT7A, of which the expression occurs throughout the entire MD [18]. Only after birth, *Wnt7a* expression is restricted to the epithelium of the oviduct and uterus. Loss of *Wnt7a* in the differentiating MD results in partial homeotic transformation of oviduct to uterus and of uterus to vagina [18,19] (Table 1). During morphogenesis, HOXA10 is not only required for proper patterning of the FRT as mutants show homeotic transformation of part (1/4th) of the proximal uterus into oviduct with fallopian tube morphology, but also later in the mature uterus for successful embryo implantation. *Hoxa10* null female mice ovulate normally, but are infertile due to (pre-)implantation defects, showing impaired decidualization [20]. In *Hoxa11* deficient mice, the uterus is thinner and shorter than normal and endometrial glands are lacking, suggesting a partial homeotic transformation to oviduct [23]. HOXA13 is required for correct MD formation and differentiation; *Hoxa13* null mutations in mice are embryonically lethal, while female embryos miss the caudal part of the MD [24]. In addition to mutations in *Hoxa10*, alterations in *Hoxa13* paralogous genes (such as *Hoxd13*) also show defective morphogenesis of the MD. Additionally, the TF p63, encoded by *Tp63* and expressed in cervicovaginal epithelium, plays an important role in FRT development. In *Tp63* null mice, the cervicovaginal epithelium contains columnar-like uterine epithelial cells, demonstrating the decisive role of *Tp63* in MD epithelium differentiation to diverge between the cervicovaginal and uterine epithelial phenotype [26].

The FRT organs are still relatively immature at birth and undergo further development during prepuberty. Regarding the mammalian uterus, three events take place: (i) differentiation and growth of the myometrium (i.e., muscle layer under the endometrium; Figure 1 and Figure 2); (ii) organization and stratification of the endometrial stroma; and (iii) development of endometrial glands (adenogenesis) [43]. During the latter process, glands establish by budding and differentiation of precursor luminal epithelium (LE) into glandular epithelium (GE). The forming glands elongate into the uterine stroma where they coil and branch toward the myometrium [43].

Several growth and signaling factors play a key role in the postnatal maturation of endometrial glands. Conditional knockout of *Wnt4* leads to a pseudostratified LE and decreased numbers of glands [29]. *Wnt5a* null female mice show only short and coiled uterine horns [30]. When fetal *Wnt5a* null FRT was xenografted into adult mice, postnatal development was normal, except for the absence of endometrial glands. WNT7A is also essential for gland formation, as *Wnt7a* mutant mice do not have endometrial glands [18]. Defective uterine gland development is also observed in mutants affecting the WNT downstream pathway, in particular the canonical WNT signal transmitter ß-catenin (cadherin-associated protein ß1, *Ctnnb1*) and the mediating transcription factor lymphoid enhancer binding factor-1 (LEF1). Conditional and null deletion of *Ctnnb1* and *Lef1*, respectively, abrogates endometrial gland development in female mice [31,32,33]. Adenogenesis, in parallel, requires remodeling of the endometrial tissue. Matrix-metalloproteinases (MMPs) and their tissue inhibitors (TIMPs) are involved in the needed, well-controlled ECM breakdown to allow gland branching. Several MMPs are highly expressed in the developing uterus and *Timp1* knockout mice exhibit an increased number and accelerated formation of glands [34,35]. Endometrial gland formation is controlled by paracrine and autocrine signaling between the epithelial and stromal cells, mediated by fibroblast growth factors (FGFs), such as FGF7 and FGF10, and hepatocyte growth factor (HGF). Insulin-like growth factor 1 (IGF1) is also associated with endometrial morphogenesis and *Igf1* null mutant female mice show a hypoplastic uterus and fail to ovulate, resulting in infertility [36]. Forkhead box (FOX) transcription factors are also involved in postnatal FRT maturation. Conditional deletion of *Foxa2* leads to inhibition of gland differentiation and resultant infertility [37]. In mice, *Foxl2* is expressed in the differentiated stromal layer, and conditional deletion in the postnatal uterus results in infertility, showing reduced thickness of the stroma and a hypertrophic, disorganized myometrial layer [38]. The distal-less homeobox (*Dlx*) genes 5 and 6 are expressed in the LE and the developing GE in the postnatal mouse uterus [39]. Conditional deletion of both genes results in abnormal gland development in neonatal mice, and eventually complete infertility. Estrogen receptor α (ERα) is also essential for uterine growth and development after birth. *Esr1* null mutant mice display hypoplastic uteri with reduced endometrial epithelial cell proliferation and gland numbers compared to wildtype mice [40,41]. Thus, *Esr1* is not essential for uterus organogenesis, but indispensable for normal postnatal uterine gland development and growth.

Relatively few genes have been identified that regulate MD formation in humans. Most knowledge arose from women with uterine abnormalities which were genetically screened. Specific syndromes such as Mayer–Rokitansky–Küster–Hauser (MRKH) and hand-foot-genital (HFG) syndrome [44], associated with FRT defects including MD aplasia, MD persistence, and MD fusion and patterning defects, have been well studied [44]. MRKH is characterized by an underdeveloped uterus and absence of the upper vagina. Multiple genes have been associated with MRKH including *WNT4*, transcription factor 2 (*TCF2*, also known as *HNF1**β*) and *LIM1* (*LHX1*) (Table 1) [44]. Patients harboring *WNT4* loss-of-function mutations show the absence of the uterus and upper vagina, excess androgens and associated masculinization [15,16]. Certain MRKH cases are characterized by deletions of the chromosomal region 17q12 that contains both the *TCF2* and *LHX1* loci [44]. Heterozygous mutations in *TCF2* result in serious FRT abnormalities, such as the presence of a double uterus with two separate cervices (uterus didelphys) [28]. Sequence analysis of *LHX1* in multiple MRKH patients revealed five heterozygous mutations associated with the disease [13,14]. A genome-wide association study (GWAS) found that two single nucleotide polymorphisms (SNPs) in *WNT9B* were significantly associated with MRKH syndrome [17]. Sequencing also revealed SNPs in the *TP63* gene among women with MD anomalies [27]. In HFG syndrome, there are defects in MD fusion and patterning. The FRT defects range from longitudinal or double vagina to double uterus and cervix [44]. Mutations in *HOXA13* have been shown to cause HFG syndrome, displaying incomplete MD differentiation resulting in partial fusion [25] (Table 1). Like in rodents, *HOXA10* is also required for correct MD fusion in humans, and mutations result in uterine malformations such as uterus didelphys [21,22]. Nonetheless, genetic causes of most FRT anomalies in humans remain unknown, and much more research is needed to identify molecular and cellular players in MD and FRT formation and differentiation. Of note, multiple developmental genes expressed in the FRT are overexpressed in gynecological cancers [45]. Current research in the field of human FRT anomalies is mainly limited to sequencing efforts and thus far failed to decipher the nature and disease-causing pathways. Hence, development and application of new human research models are considered of high value to generate scientific and clinical progress in this field.

### 2.2. Endometrium Biology

#### 2.2.1. Endometrium Remodeling during Menstrual Cycle

Histologically and functionally, the human endometrium consists of two layers, the upper *functionalis* and lower *basalis* layer [46] (Figure 1). The *functionalis* layer, proximal to and lining the uterine lumen, contains the glands embedded in supportive stroma. The *basalis* layer contacts the myometrium and consists of branching glands and their endings, dense stroma and large blood vessels. Endometrial remodeling during the menstrual cycle is tightly regulated by ovarian estrogen and progesterone, and the hormonal interplay during the proliferative and secretory phase coordinates its growth and differentiation, respectively [46].

Immediately in parallel with the shedding process of major parts of the *functionalis* layer (menses), re-epithelialization is initiated. The endometrial repair process is completed within 48 h of onset of shedding. During the proliferative phase, estrogen levels increase, together with the expression of its receptor Erα, as well as the progesterone receptor (PR) [46]. This phase is characterized, in addition to expansion of stromal cells, by re-epithelialization through extensive proliferation of (residual) glandular epithelial cells, resulting in rapid thickening of the tissue (to ~7 mm) and restoration of the LE [47]. The peak of estrogen not only ensures GE proliferation, but also angiogenesis to provide sufficient oxygen and nutrients to support the tissue’s expansion, and the growth of supportive stromal cells [46]. The re-formation of the *functionalis* layer is completed at the end of the proliferative phase. Increasing levels of progesterone mark the start of the secretory phase in which the endometrium further expands to a size of ~10 mm. Under the influence of progesterone, decidualization occurs, through which stromal cells differentiate into large rounded decidual cells with enhanced secretory function. The epithelial cells also undergo morphological changes, thereby increasing ciliogenesis and secretory activity (Figure 1) [48]. Production of the many secretory molecules serves to prepare the endometrium for blastocyst attachment, implantation and earliest development [49]. If no blastocyst interaction and implantation occurs, estrogen and progesterone levels drop again, which triggers menstruation, with endometrial cell apoptosis and ECM remodeling, thereby degenerating large parts of the *functionalis* layer which is shed into the uterine cavity [46] (Figure 1). This repetitive and long-term cycling of growth and regression epitomizes the remarkable plasticity of the endometrium. Remodeling processes in tissues are generally associated with the presence and activity of tissue-resident stem cells. Hence, it is logical to propose that the endometrium also houses stem cells which are involved in the dynamic regeneration. Stem cells are hypothesized to be located in the *basalis* layer, which remains after menstruation and thus may support the reconstitution of the stromal and epithelial cell compartments (Figure 1) [50,51,52,53].

#### 2.2.2. Endometrium Stem Cells

Proficient tissue renewal typically implies the presence of stem cells (such as in skin and intestine; [54]). However, the presence and identity of stem cells in the endometrium and their involvement in endometrium physiology remain far from settled. Although actively searched after, no conclusive evidence has been reached yet on whether stem cells drive the regeneration and proliferation after menstruation and/or during the proliferative phase. Moreover, one should distinguish between epithelial and stromal stem cell populations, unless a bipotent cell is present that gives rise to both compartments. Stem cells are typically embedded in a specialized microenvironment (i.e., stem cell niche) with regulatory and supportive cells that protect the stem cells and regulate their maintenance (self-renewal), progression and expansion as transit-amplifying progenitor cells, and differentiation toward mature tissue cells, all through specific ECM and paracrine signals [55,56]. Stem cell niches have not yet been conclusively defined in the endometrium. The *basalis* layer may house such niches, but no definite physical configurations have been pinpointed yet [50,51,52,53].

In mice, genetic tracing studies provided evidence for endometrial stem cell populations. Lineage tracing of the WNT target gene *Axin2* revealed self-renewal and multipotency of AXIN2-expressing (AXIN2^+^) cells during endometrium development, growth and regeneration, together pointing to the presence of long-lived bipotent epithelial stem cells, giving rise to ciliated and non-ciliated/secretory cells of both GE and LE in the long run [57]. It was found that the AXIN2^+^ cells reside in the endometrial glands at their ‘base’, although not clearly specified [57]. However, the AXIN2^+^ GE cells did only minimally contribute to the LE, and may thus rather represent a GE-specific stem/progenitor cell. Another WNT-related factor, leucine-rich repeat-containing G-protein coupled receptor 5 (LGR5), is broadly expressed in uterine epithelial cells during embryogenesis, and becomes confined to the tip of developing uterine glands after birth [58]. Lineage tracing supported a crucial role of these LGR5^+^ stem/progenitor cells in endometrial gland development, embodying the WNT-driven gland formation in the developing uterus.

In humans, over the past years, several populations of endometrial stem/progenitor cells have been proposed (reviewed in [53]). Endometrial mesenchymal stem cells (MSC) were isolated as CD140b^+^CD146^+^ or SUSD2^+^ cells and were shown to be self-renewing and multipotent (i.e., having the ability to differentiate into mesodermal lineages). The cells, which may represent the stem cells of the stromal compartment, were found to reside in both the *basalis* and *functionalis* layer as perivascular cells [47,59]. Candidate endometrial epithelial stem/progenitor cells expressing N-cadherin were found to reside in the *basalis* layer at the bases of the glands. These cells appear to differentiate in a hierarchical spatial pattern as they acquire temporary expression of different other stemness markers, including combinations of SSEA1, SOX9, AXIN2 and ALDHAH1, when located nearer to the *functionalis* layer. Cells that were FACS-isolated using these markers showed clonogenic capacity, and higher self-renewal and proliferative activity than the non-expressing cells [60,61]. In another study, LGR5-expressing cells were found in the luminal epithelium and in the basalis layer [62]. The authors hypothesize the presence of two epithelial stem/progenitor cell pools; one residing in the basalis (SSEA-1^++^SOX9^++^LGR5^+^) supporting the regeneration of the functionalis after menstruation, while the other population (LGR5^++^SSEA-1^+^SOX9^+^) is needed to support the embryo-implantation process, and for the maintenance of luminal epithelial cells [62]. FACS-isolated ‘side population’ cells of epithelial and stromal origin could form endometrial-like structures when transplanted under the mouse kidney capsule [63]. Finally, there are also indications that bone marrow-derived stem cells end up in the endometrium where they contribute, although limitedly, to the tissue’s remodeling [59]. Apart from (or in addition to) stem cells projected to underlie the dynamic endometrial tissue regeneration, alternative explanations may exist, such as proliferation of differentiated cells (potentially after prior dedifferentiation) [64], or MET, through which re-epithelialization starts from stromal (stem) cells (suggested, among others, by genetic mapping of endometrial mesenchymal stem cells [65]), or a combination of several different mechanisms.

Taken together, despite extensive research, no definitive conclusions can be drawn at present about the identity and role of endometrial stem cells. Interestingly, the endometrium is being increasingly subjected to granular mapping through single-cell (sc) multi-omics profiling. Such endeavors are expected to substantially aid in the identification and understanding of endometrial stem cells.

#### 2.2.3. Unraveling the Endometrium Make-Up by Single-Cell Transcriptomics

Sc transcriptomics is a powerful tool to define tissue cellular composition by mapping the different cell type (sub-)populations, while additionally having the potential to uncover new or rare cell types or cell states. Moreover, it allows the study of complex cellular events and interactions as they occur in a dynamic microenvironment. Here, we give a brief overview of recent sc profiling endeavors of the human endometrium, including the (as yet) minimal information extracted on potential stem/progenitor cells (Table 2).

A recent study profiled the human endometrium by sc RNA-sequencing (scRNA-seq) analysis across all phases of the menstrual cycle [77]. The cells were largely clustered in seven groups (see Table 2). The major phases of endometrial transformation during the menstrual cycle could be identified through unbiased analysis of the acquired sc transcriptomes. Enrichment for genes that changed across the menstrual cycle revealed that the WOI opens with an abrupt and discontinuous transcriptomic activation of multiple genes in the unciliated epithelium (including known markers such as *PAEP*, *GPX3* and *CXCL14*). In parallel, the stromal cells show decidualized characteristics, although in a more gradual and continuous transition (with, among others, expression of *DKK1*, *CRYAB*, *IL15* and *FOXO1*). In both cell compartments, the WOI closes with gradual transcriptomic changes. New markers were identified to delineate the ciliated epithelial subset (e.g., *C11orf88*, *FAM183A*, *CDHR3*). The study also advanced new transcriptomic signatures of LE and GE (e.g., *WNT7A* overexpression in LE during the proliferative phases, and differential expression of *MMP26*, *LIF* and *MT1E*), and provided evidence for a direct interplay between stromal fibroblasts and lymphocytes during decidualization. Together, this study represents a milestone in endometrium sc profiling by providing a molecular and cellular cartography of the changes that occur in the human endometrium across the menstrual cycle. Of note, although the topic of stem cells was not highly elaborated, expression of *PDGFRB*, *MCAM* and *SUSD2*, previously identified in endometrial cells with mesenchymal stem cell characteristics, was found in the smooth muscle cell cluster (thus pointing to putative stromal stem/progenitor cells) (Table 2). Another study [70] found expression of these markers in all cells of the perivascular cell cluster (Table 2). Further investigation is needed here, especially since stromal progenitor cells and pericytes are typically isolated using similar cell-surface markers (such as THY1/CD90). The latter study also revealed multiple stromal cell populations suggestive of specific stromal niches that may differentially control local processes such as inflammation and ECM composition. Along the same line, stromal cells of the peri-implantation endometrium were subclustered into decidual cells and senescent decidual cells (Table 2), marked by *SCARA5* and *DIO2*, respectively [67].

Another recent study characterized the human endometrial cells in both time and space by applying scRNA-seq, as well as spatial transcriptomics on endometrial biopsies of the proliferative and secretory phases [72]. The study identified 14 cell clusters that could be grouped into five major classes (Table 2). The epithelial cell population was further looked at in depth, revealing the presence of stemness markers SOX9 and LGR5. By inferring ligand–receptor interactions between cell types through application of CellPhoneDB [78], it was deduced that WNT and NOTCH signaling may play distinct but complementary roles in the development/differentiation of ciliated epithelial cells (more present in the LE) and secretory cells (more in the GE). In particular, dominant WNT over NOTCH signaling leads to differentiation into the ciliated cell lineage, while predominant NOTCH over WNT generates the secretory cell lineage (Figure 1); both conclusions directly linked to spatial expression patterns of the pathways as exposed by spatial transcriptomics. Importantly, this proposed regulation was functionally validated using organoid models (see below) through WNT or NOTCH pathway activation and inhibition, and was found to be dependent on the presence of estrogen and progesterone. This study represents an important step forward in understanding the epithelial compartment, both transcriptionally and spatially. Of note, spatial transcriptomics has great potential to locate possible stem cell niches, although in this study it is not advanced yet.

The sc profiling studies also revealed an important immune cell fraction in the endometrium. The leukocyte population includes T and B cells, natural killer (NK) cells, macrophages, dendritic cells, neutrophils and mast cells [72,77,79]. During the menstrual cycle, there are important alterations in their number and type [80]. As one example, uterine NK (uNK) cell number largely increases in the late secretory phase and further expands during early pregnancy. Uterine NK cells are the main type of immune cell in the decidualized endometrium and play an essential role toward pregnancy. Acute cellular senescence of stromal cells drives the transient inflammatory decidual response with tissue remodeling, needed for endometrial receptivity [71,81]. The uNK cells target and clear the senescent decidual cells at the right time and level. If not occurring in an appropriate manner, endometrial receptivity is disturbed and pregnancy cannot unfold. Indeed, recurrent pregnancy loss is characterized by uNK cell deficiency in parallel with a shift from a majority of decidual to senescent decidual cells [67]. Increased cellular senescence in epithelial and stromal cells, together with high collagen deposition and decreased proliferation (as based on downregulation of proliferative gene expression), is found by scRNA-seq analysis in ‘thin endometrium’ versus healthy endometrium, anomalies that likely underly the infertility associated with this pathology [73]. In addition, fewer macrophages and uNK cells are present and aberrant activation of multiple pathways (such as the epidermal growth factor (EGF) pathway) was observed. Uterine NK cells also increase the blood flow at the fetus–maternal interface by secreting angiogenic factors such as vascular endothelial growth factor (VEGF), and help the invasion and migration of trophoblast cells by secreting cytokines and growth factors [82], such as leukemia-inhibitory factor (LIF)), that promote trophoblast cell invasion through interaction with trophoblast receptors [80]. In addition to uNK cells, regulatory T (Treg) cells are recruited to the decidualized endometrium where they play a key role in immune tolerance of the fetus [83]. Imbalance of immune cells has been associated with obstetric complications. Insufficiency in Treg cells in the decidualized endometrium is correlated with recurrent pregnancy loss, preterm births and preeclampsia [83,84]. Increased numbers of uNK cells are observed in women with spontaneous abortion and recurrent pregnancy loss [85,86,87,88]. Further investigation of the immune system in the endometrium by deeply diving into sc transcriptomes, among other methods, is needed to increase our understanding of the contribution of immune cells to endometrium physiology, fertility and pregnancy, which eventually could lead to immune biomarkers for poor reproductive health, as well as to therapeutic targets.

## 3. Modeling Endometrium Development and Biology

Molecular and cellular mechanisms underlying endometrium development and biology (cyclic remodeling, receptivity) in humans are far from clarified. Lack of representative and reliable study models remains the most important hurdle. Presently used animal models are not sufficiently translatable as they differ in key aspects of endometrium development, function and regulation. As a few examples, decidualization in rodents is triggered by the presence of the fertilized egg, whereas this process in humans unfolds automatically every month [89]; moreover, progesterone receptor A (PR-A) plays a dominant role during decidualization in mice, whereas progesterone receptor B (PR-B) is more important in humans [90].

### 3.1. Modeling Endometrium Development

Modeling (recapitulating) endometrium development in vitro is still very limited. Induced pluripotent stem cells (iPSCs), although extensively used to sculpture development of other organs, has only scarcely been applied to model FRT and derived endometrium organogenesis. Development of the MD has been achieved by driving human iPSCs consecutively through a primitive streak, intermediate mesoderm and coelomic epithelium stage [91], thereby mimicking the in vivo sequential process where invagination of coelomic epithelium eventually forms the MD. Following modulation of WNT and BMP signaling giving rise to MD cells, subsequent exposure to MD growth factors (i.e., WNT4 followed by follistatin) promoted the establishment of fallopian tube epithelium precursors [91]. Regarding the endometrium, iPSCs, following progression through the different stages from the primitive streak to MD, have been differentiated into endometrial stromal fibroblasts [92]. The cells expressed the endometrial (stromal) markers HOXA10, HOXA11 and PR, and showed a decidualization response to hormonal stimulation [92]. Moreover, iPSC-derived endometrial stromal fibroblasts were also generated in another study [93], applying a comparable approach by inducing the intermediate mesoderm through WNT activation, followed by sequential WNT and BMP signaling activation to achieve an endometrial stromal fibroblast phenotype. To our knowledge, endometrial epithelial cells have not yet been derived from iPSCs in vitro, for which still other specific signaling cues toward directed differentiation will be needed. Of note, the studies mentioned did not touch upon, or reveal clues on, the identity of endometrium stromal stem cells.

### 3.2. Modeling Endometrium Biology

#### 3.2.1. In 2D

Conventional 2D cultures have been extensively used to study endometrium biology in vitro, and have advanced our understanding of cellular responses to biophysical and biochemical stimulations such as decidual alterations. Although primary endometrial stromal cells can be expanded long-term in these conditions, they eventually lose phenotype and hormone responsiveness. Primary epithelial cells from the endometrium do not expand in 2D monolayer culture, and also quickly lose their physiological hormone sensitivity. Immortal(-ized) cell lines (such as Ishikawa cells [94]), which are frequently used, are karyotypically and physiologically abnormal. Moreover, a 2D monolayer lacks the in vivo 3D physiology and complexity. To better approach the multicellular composition of the endometrium, models have been established by co-culturing different endometrial cell types, often using an ECM-mimicking gel to embed and support the cells ([95,96]; reviewed in [97]). These models have also been applied to study the interaction with the embryo (reviewed in [98]; see further below). Although 2D cell cultures have dominated the field for a long time, recent research has shifted toward culturing three-dimensional (3D) structures as they provide enhanced biomimicry in terms of structure and physiology.

#### 3.2.2. In 3D

A new and powerful in vitro 3D model to study human tissue biology (as well as pathology) is provided by the organoid technology. Organoids can be established from tissue-specific stem cells or from PSCs (i.e., embryonic stem cells (ESCs) and iPSCs) [99,100]. Organoids from tissue-specific stem cells are defined as 3D cell structures that develop through self-renewal, proliferation and self-organization of the stem cells under specified culture conditions. In particular, the organoids arise when embedded in an ECM scaffold (such as Matrigel) under WNT signaling activation [101]. The organoids recapitulate phenotypical and functional characteristics of the native tissue’s epithelium, and are expandable long-term, while robustly retaining (patho-)biological properties and remaining genomically stable [99]. Hence, organoid models overcome the hurdle of the usually limited availability and expandability of primary human tissues (as obtained from clinical samples), hence being highly instrumental and essential for thorough mechanistic and translational scrutiny. Moreover, organoids can be cryopreserved and biobanked, enabling not only the study of specific diseases in different forms, but also providing an accessible screening platform for (new) drugs, which may eventually enable individually tailored therapies.

The first successful tissue stem cell-derived organoid model was established from mouse intestine. The organoids were developed from LGR5-expressing intestinal stem cells present in the tissue crypts [102]. Intestinal fragments or (stem) cells were embedded in Matrigel and cultured in a cocktail of growth and signaling factors that mimicked the intestinal stem cell niche, in particular containing EGF and WNT activators (WNT3A and the WNT-amplifying LGR5 ligand R-spondin1 (RSPO1)) to maintain and proliferatively stimulate the stem cells, and the BMP signaling inhibitor Noggin to prevent the stem cells’ differentiation. These components form the basis of generic organoid culture medium, which is, according to the specific tissue that is modeled, further enriched with factors that play in the tissue-specific stem cell niche or, if the niche is not defined, that recapitulate the embryogenetic process of the tissue under study. Since the pioneering study in 2009, organoid models have been developed from manifold tissues of both mouse and human origin [103]. Despite their great power, organoids have the specific limitation that they only recapitulate the epithelial (stem cell) compartment of the tissue. Therefore, to advance the model to closer mimic the tissue, composite systems are being designed in which organoids or (organoid-derived) epithelial cells are combined with other cell types of the tissue (mesenchymal, immune, endothelial cells) [104] (see below).

Organoids can also be established from PSCs by recapitulating the embryogenetic program of the targeted organ to achieve the appropriate lineage specifications. In addition to FGF, WNT and BMP signaling, which are important to establish endodermal, ectodermal and mesodermal lineages, other specific signaling factors are added in a strictly time-controlled manner to obtain the specific cell types that compose the desired organ [103]. When proper lineage specification is achieved, cells are cultured in 3D, either by aggregation or by embedding into an ECM surrogate such as Matrigel [103], e.g., gastric organoids [105]. By mimicking the developmental process, several cell types of the organ are generated simultaneously, which is an advantage compared to the pure epithelial organoids established from tissue stem cells. However, the iPSC-derived 3D cell mixture may contain aberrant cell types (e.g., PSCs that did not differentiate or off-target cell types that formed). Another shortcoming of organoids derived from PSCs is that they remain mainly trapped in an immature phenotype [106], and that the 3D structures are not multipliable in contrast to tissue-derived organoids. Importantly, both organoid models can be harnessed for cutting-edge experimental manipulations. In particular, CRISPR/Cas9-mediated gene editing can be applied, which is highly instrumental to decipher mechanisms of tissue development, physiology and pathogenesis using the organoids as close avatars.

Five years ago, organoids were also successfully developed from endometria of both mouse and human origin [1,2]. The organoids were found to reproduce key biological features of the endometrial epithelium, including glandular-type organization and expression of specific markers (e.g., FOXA2, PAX8). Moreover, the organoids displayed physiological responsiveness to the endometrium’s core regulatory hormones, estrogen and progesterone, resulting in the replication of the menstrual cycle in a dish [1]. Mimicking the in vivo hormonal exposure regimen reproduced first the proliferative phase with increased cell proliferation and expression of specific markers (such as TRH), followed by the secretory phase with morphogenetic processes (formation of columnar epithelium with subnuclear vacuolation, increased glandular tortuosity and ciliogenesis), and specific expression alterations with mucus production, downregulation of ERα and upregulation of specific markers such as PAEP [1]. Finally, hormone withdrawal induced the menstrual phase in which dying cells were shed by the organoids. Hence, the human endometrial organoids provide a powerful tool to study the mechanisms underlying endometrial biology, i.e., the cyclic remodeling of endometrial epithelium. In addition, deregulation of the remodeling activity which may lead to endometrial dysfunction can also be studied, not only disease-wise (such as in endometriosis; see below), but also gene-wise. As an example, *Dlx5/6* has been found expressed in the GE of an adult human endometrium. Based on findings in mice, in which conditional deletion leads to abnormal gland development and infertility [39], their role in human endometrial gland biology can be studied through CRISPR/Cas9-targeted mutation in organoids. Moreover, mutating disease-associated genes (see Table 1) in organoids (or use of patient-derived organoids) may shed light into endometrium-impacting pathways in syndromes such as MRKH to decipher both the nature and heterogeneity of the disease at the level of the endometrium. Meanwhile, organoids have not only been established from endometrial tissue at different stages (proliferative, secretory) of the menstrual cycle [1,2,66,72], but also from decidualized and postmenopausal endometrium [2,107], and even from menstrual flow, resulting in organoids that are transcriptionally identical to primary endometrial epithelium [108].

Interestingly, endometrium-derived organoids are increasingly subjected to scRNA-seq profiling (Table 2). The organoids were found to be heterogeneous in cell composition and to contain ciliated and secretory cells [68], as well as different hormone-responsive subpopulations [66]. Estrogen-treated organoids (mimicking the proliferative phase) were made up of five major cell types (distributed over 13 subclusters), including ‘stem cells’ (Table 2), although these were not further elaborated on. When the organoids were treated with estrogen and progesterone (mimicking the secretory phase), an additional cell cluster emerged consisting of secretory cells. In addition, specific gene expression changes were observed in differentiation markers such as PAEP and SPP1. Additionally, the number of ciliated cells was found to increase upon exposure to estrogen, while the number of ‘stem cells’ decreased [66]. Estrogen signaling appeared to be the main driver of ciliogenesis, which was also promoted by NOTCH signaling inhibition [68,107], actually in reciprocal interplay with WNT activation [72]. Inhibiting NOTCH while activating WNT in organoids shifts the balance towards ciliated cell differentiation, whereas the reverse interference towards secretory cell differentiation [72] (Figure 1), at the same time suggesting the existence of one and the same stem/progenitor cell type for both epithelial lineages. Extensive sc profiling of organoids and comparison with sc transcriptomes and gene expression programs of ex vivo endometrium (epithelial) cells are now needed to deeply benchmark the organoid model, at the same time providing a powerful resource to dissect and decipher endometrium biology.

An important implication of the fact that organoids typically grow out from the tissue’s stem cells is that endometrium-derived organoids can help to uncover and conclusively identify the endometrium epithelial stem cells. Indeed, mouse endometrial AXIN2^+^ cells were found to form (expandable) organoids, whereas the remaining cells did not [57]. Very intriguingly, in humans, organoids do not establish from SSEA-1^+^ cells [2]. However, spontaneous formation of gland-like spheroids [60] and immature gland structures [109] has been observed from SSEA-1^+^ cell populations, thus supporting their progenitor phenotype, with SSEA-1 being generally regarded as a stem cell marker in the human endometrium [53]. Recent scRNA-seq profiling discovered that the largest cell population in endometrial organoids mapped onto the *MUC5B^+^* epithelial cells from primary tissue, potentially revealing a new candidate endometrial stem/progenitor cell population [76] (Table 2). Menstrual flow-derived organoids display a transcriptional signature rich in stem/progenitor cell markers [108], and proliferative SOX9^+^LGR5^+^ cells spatially map onto the suggested regenerating cells of the proliferative (estrogen-dependent) phase within the hypothesized stem cell niche [72]. Together, these organoid findings strengthen the concept that organoids can serve as a powerful model to study endometrium physiology and as a potential tool to conclusively define the endometrium epithelial stem cells which may underlie the impressive regenerative capacity of the tissue. CRISPR/Cas9-targeted mutations in stem cell regulatory genes may then support their stem/progenitor cell role.

In a further interesting advancement, the organoid model was turned into a human in vitro embryo implantation model. The closed 3D structure was opened up into an ‘open-faced endometrial layer’ (OFEL) to expose the apical epithelium side to the embryo [110]. Stem cell-derived blastocyst models (blastoids) were found to attach to the OFEL, although only when the latter was hormonally primed to reach the WOI. Attachment was abrogated by the contraceptive levonorgestrel. Interestingly, physiological events of embryo attachment were recapitulated, including production of pregnancy hormone, directional attachment through the polar trophectoderm, specific gene expression alterations and morphogenetic events (e.g., repulsion of the endometrial cells and first steps of embryo morphogenesis) [110]. Thus, the endometrial organoid technology is highly valuable (and essential) to develop a human in vitro implantation model which can now be used to decipher the endometrium’s role and biology at implantation, and the mechanisms of embryo–endometrium interplay, otherwise strongly hidden from exploration because of occurring inside the womb.

#### 3.2.3. Advanced Composite Endometrium Models

The endometrium contains several cell types including epithelial, stromal, immune and endothelial cells (Figure 1). Hence, epithelial organoids should be topped up with other endometrial cell types to closer mirror the tissue.

In recent years, efforts have been made to co-culture endometrial epithelial organoids with endometrial stromal cells (Figure 3). Since organoid culture conditions favor growth of epithelial cells (because of the presence of ECM anchoring and epithelial growth factors), endometrial stromal cells are swiftly lost during organoid culture and passaging. Therefore, a co-culture system in which both different cell types are thriving is not straightforward. When epithelial and stromal cells are seeded in an agarose mold, they self-aggregate and form a 3D structure with a stromal cell center and an epithelial cell outer layer [111,112] (Figure 3). The stromal cells provide the scaffold that support the epithelial cells, and these so-called ‘ECM scaffold-free organoids’ (although not adhering to the currently used definition of organoids as described above) were found responsive to sex steroids. Another study used a porous collagen scaffold in which stromal cells were able to proliferate, and the organoid-derived epithelial cells which were seeded on top formed a luminal-like epithelial layer (Figure 3). Both cell types were functionally responsive to sex hormones and produced their own matrix proteins [113]. In another approach, human PSC-derived endometrial stromal fibroblasts were co-cultured with decidua-derived endometrial organoids (i.e., from term placenta) to investigate epithelial–stromal signaling [93]. When both cell types were seeded in a Matrigel droplet, the stromal fibroblasts self-aggregated and interacted with endometrial organoids to form a composite structure with an inner epithelial cell layer and an outer stromal zone, responding to hormone exposure. However, substituting the PSC-derived stromal cells for their primary tissue counterparts did not result in similar successful co-culture. In contrast, Rawlings and colleagues recently succeeded in combining primary stromal cells with epithelial organoids, thereby forming ‘assembloids’ [71] (Figure 3). Dispersed organoids and primary stromal cells were co-seeded in a collagen hydrogel, resulting in the formation of gland-like organoids surrounded by a matrix rich in stromal cells, thereby resembling the architecture of native endometrium [71]. Interestingly, the presence of stromal cells abrogated the need for adding certain compounds to achieve epithelial organoid differentiation, showing the importance of stromal cells in providing a decidualization niche for the epithelial cells. scRNA-seq analysis of assembloids showed a divergence of epithelial and stromal cells toward differentiated and senescent subpopulations when hormonally treated to mimic the secretory phase (Table 2). Intriguingly, injection of a human blastocyst into the composite 3D structure after hormonal decidualization priming led to a certain level of embryo development and movement. Chemical elimination of the senescent decidual cells restricted these embryo processes, thereby supporting a key role of senescent decidual cells in successful implantation (see above).

Further advancing the endometrial mimics will require the addition of immune cells, in particular uNK cells and macrophages which play an important role in endometrium receptivity and regeneration, respectively, and of endothelial cells (e.g., endometrium- or iPSC-derived [114]). In this context, vascularization may also help to further mature the organ mimic. Such assembloids have already been developed from other organs such as the bladder, including epithelial, stromal, endothelial and immune cells, as well as a muscle layer [115]. This endeavor to achieve still more complex biomimetic designs will strongly benefit from the unravelment of the endometrium cellular make-up and the identification of its stem cells, in efforts currently being conducted through sc multi-omics.

Microfluidic technology may offer an alternative or additional trick to enhance the mimicry of complex tissues and environments. A microphysiological system of the endometrium has been built in which endometrial stromal cells are co-cultured with endothelial cells to evaluate paracrine and endocrine crosstalk [116]. For instance, it was shown that endothelial cells, when placed under shear stress, secreted prostaglandin which led to a significant enhancement of the stromal decidualization response. Moreover, such next-generation models would not only be instrumental to decode the endometrium biology (including critical regulators of implantation), but also pathology. Indeed, accurate interaction between all players in the endometrium, i.e., epithelial (glands), stromal, vascular and immune cells, is essential to coordinate a proper function. Imbalance can lead to pregnancy failure and infertility, as well as to benign and malignant diseases.

Organoids are also amenable to study the interactions of a tissue with natural or pathogenic microorganisms [117]. The endometrial microbiome remains underexplored when compared to, for instance, the vaginal microbiome of which an imbalance is well-known to lead to pathologies. The endometrial microbiome is not yet highly defined regarding its impact on endometrial biology (cycling, receptivity). To tackle this gap, endometrial organoids could be co-cultured with bacteria from the tissue’s microbiome (e.g., *Lactobacillus* species) and the impact on biology assessed, which would enhance our understanding of the importance of the microbiome in endometrial functioning. Moreover, the impact of pathogenic microorganisms can also be studied. For instance, infection of mouse endometrium-derived organoids with *Chlamydia trachomatis* [118] revealed bacterial inclusions within the cellular cytosol where the pathogen underwent a full developmental cycle. This model complements a human fallopian tube organoid *Chlamydia* infection model [119], in which the infected organoids exhibited a less differentiated phenotype with increased stemness characteristics and hypermethylation of DNA.

Finally, to construct ‘a uterus in a dish’, at least comprising endometrium and myometrium mimics, not only the different cell types need to be included, but also the complex architectural organization needs to be recapitulated, such as *basalis* and *functionalis* layers with differing gland configurations [52,120], and stem/progenitor cell hierarchies [121]. Eventual faithful modeling will strongly help to enhance our understanding of uterus biology and pathology. It would also increase the efficacy of preclinical drug discovery and toxicity testing (already possible with organoids), and be a useful clinical tool towards optimal and targeted management of individual patients with uterus anomalies, and of endometrium-based infertility patients.

## 4. Modeling of Endometriosis

### 4.1. Endometriosis

Endometriosis is a leading cause of female morbidity, affecting 1 in 10 women of reproductive age [122,123]. The disease is defined by out-of-the-uterus (ectopic) growth of endometrial-like tissue, and is associated with severe symptoms such as debilitating chronic pelvic pain and dyspareunia, as well as strongly reduced fertility to infertility (in 30–50% of the patients) [122], all together causing a profound negative impact on quality of life. Endometriosis encompasses three major lesion types, i.e., peritoneal superficial, ovarian cystic (endometrioma) and deep infiltrating; and four different stages based on the American Society for Reproductive Medicine (ASRM) scoring system assessing type, localization and size of the lesions and their associated adhesions [124]. Superficial peritoneal lesions are typical for stage I–II, whereas deep infiltrating endometriosis and ovarian endometriomas are usually described in stage III–IV. Intriguingly, clinical symptoms do not highly correlate with stage.

Although surgical (laparoscopic) resection of the endometriotic lesions can improve pain and infertility, long-term benefits remain unclear in terms of symptom management and clinical outcome, showing frequent recurrence of lesions and infertility [122]. Pharmacological treatment involves suppression of estrogen activity (which drives the lesions in their growth behaviour), for instance by oral contraceptives. However, there are major limitations: it is not compatible with pregnancy or infertility treatment (such as IVF); pain symptoms may become resistant during, or recur after, hormonal treatment; and side effects are multiple (e.g., uterine bleeding, weight gain, mood changes) [123]. Moreover, when treatment is stopped, the disease invariably recurs. Thus, novel therapeutic approaches are highly needed, particularly involving drugs directly targeting the endometriotic lesions without disturbing hormonal balances.

Despite the first description of endometriosis more than a century ago, little is understood about the underlying pathogenesis. Main hypotheses include retrograde menstruation, coelomic metaplasia, neonatal uterine bleeding, and lymphatic and vascular metastasis of endometrial cells [122,123]. Retrograde menstruation (known as ‘Sampson’s theory’) remains the most accepted pathogenetic mechanism, posing that a backward flux of menstrual debris, which contains endometrium epithelial and stromal cells and tissue fragments, occurs through the fallopian tubes into the peritoneal cavity where cells/fragments attach and develop into the endometriotic lesions that, similar to the endometrium, sway on the waves of the estrogen/progesterone cycling [125]. Whether the endometrial (epithelial/stromal) stem cells are the leading drivers for this ectopic outgrowth has often been postulated but is unknown as of yet, largely because the stem cells’ identity is not conclusively defined (reviewed by [126,127]). Whereas many women (~90%) experience retrograde menstruation, ‘only’ 10% develop endometriosis. Hence, a predisposing background encompassing (epi-)genetic and environmental factors is also at play. Former studies have indeed demonstrated that endometriosis has a familial nature, and GWAS have identified specific genomic regions associated with a risk of developing the disease [123]. Taken together, it is clear that a more thorough understanding of endometriosis pathogenesis is needed to open up new avenues toward diagnosis and treatment. In addition, there is not much known on how sub-/infertility is linked with endometriosis and, in particular, how the endometria of these patients differ from the endometria of healthy women. Lack of representative and reliable study models is also a major hurdle to progress in the field. Animal models currently used are not highly translatable for this specific human (and non-human primate) disease. Indeed, despite a multitude of agents reported in preclinical studies using such animal models, none have been introduced into clinical practice [128].

### 4.2. Single-Cell Profiling of Endometriosis

To gain deeper insight into endometriosis biology with the potential of revealing pathogenic and therapeutic aspects, several studies recently initiated the race of endometriosis sc profiling. 

A recent study portrayed endometriotic lesions, as well as the endometrium (biopsies), of patients (eutopic endometrium) using scRNA-seq interrogation, and identified 96 subclusters covering nine major cell types [75] (Table 2). Different expression signatures in epithelial and stromal cells were found between eutopic and ectopic tissue, suggesting transcriptional reprogramming at ectopic grafting and (out-)growth of the endometrial tissue, which may play a central role in the pathogenesis. In addition, a dysregulated involvement of immune pathways was found in endometriomas. Transcription factor gene-regulatory network (regulon) analyses put FOXO1, XBP1, MAFF and JUND regulons forward as potentially marking stromal stem/progenitor cells in the eutopic endometrium.

In a comparable study also profiling ectopic and eutopic endometrium from endometriosis patients as well as healthy endometrium [74], nine cell clusters were identified (Table 2). Interestingly, eutopic endometrium was found distinct from healthy tissue in terms of cell composition, gene expression and pathways. In particular, the eutopic tissue was found enriched in genes and terms associated with angiogenesis, ECM organization, cell motility and adhesion, all being processes proposed to play a role in endometriosis’ pathogenesis. The TGFß, MAPK, NFkB and epithelial-to-mesenchymal transition (EMT) pathways were also found to be enriched, and were therefore suggested to underlie the pathogenic process. Together, it was proposed that eutopic endometrium represents anomalies potentially leading to disease development, by defining it as a transitional state between normal endometrium and ectopic lesion. In addition, the authors identified fibroblast subpopulations which may be related to disease development. Among others, fibroblasts overexpressing steroidogenic acute regulatory protein (StAR) may be linked to the driving role of estrogen in the disease onset, propelling the cells’ proliferation, migration and invasion [74]. Furthermore, less activated T cells and fewer NK cells were detected in the ectopic lesions, which epitomizes, and may underlie, immune dysfunction, proposed to enable the outgrowth and thriving of the endometrial tissue at the foreign, ectopic location.

The most recent sc profiling study [76] performed scRNA-seq analysis of peritoneal and ovarian endometriotic lesions, which were compared to matched eutopic endometrium and healthy tissue, and also profiled derived organoids. The selected patients exhibited active disease symptoms and were under contraceptive treatment to better understand endometrial changes and endometriotic lesions independent of menstrual cycle dynamics. The study identified five major cell types, resulting in 58 subclusters (Table 2). In the peritoneal lesions, a novel and unique perivascular mural cell type was identified with a potential dual role in immune cell trafficking and angiogenesis stimulation. Moreover, the study also profiled peritoneal tissue in which the lesions were growing, and found that the microenvironment of this peritoneal niche contains immunotolerant macrophage and dendritic cell populations that display strong differences with eutopic endometrium. Importantly, similar cell type compositions were found in eutopic endometrium and peritoneal lesions, which supports Sampson’s theory that these lesions originate from eutopic endometrium. However, and intriguingly, ovarian lesions depict a distinct cell type composition and gene expression pattern compared to peritoneal lesions. Interestingly, a before non-reported progenitor-like epithelial cell subpopulation was exposed in (eutopic) endometrium and ectopic lesions, expressing MUC5B (Table 2). Interestingly, the largest cell population present in the combined organoid cohort, subjected to scRNA-seq analysis in expansion conditions, clustered together with MUC5B-expressing epithelial cells in primary tissue. SOX9 was also expressed in the MUC5B^+^ cell population; however, it was also present in other epithelial cell subtypes. Of note, Garcia-Alonso et al. showed that endometriotic peritoneal lesions are enriched in markers from the SOX9^+^LGR5^+^ epithelial cell population, such as WNT7A and KRT17, when compared to normal peritoneum. This observation is in line with an enhanced expression of SOX9 that is found in later stage endometriotic organoids [3]. Moreover, examining sc datasets from endometriosis showed enrichment in the markers (SOX9 and LGR5) of the ‘stemness’ cells that drive endometrium rebuilding during the proliferative phase, pointing to a potential disease origin by stem/progenitor cells that actively proliferate under estrogen but are not sensitive to progesterone, fitting with the typical estrogen dominance and progesterone resistance of the disease.

Taken together, these first studies profiling tissue from endometriosis patients at the sc resolution represent an important step in shedding light on the pathogenic mechanisms as well as the originating (stem) cells and the cell types discerning the patient tissue from normal (e.g., immune cells). Now, more large-scale cellular and molecular profiling efforts, associated with functional exploration (using appropriate models, see below), are needed, which will also help to find new treatment and diagnostic strategies. It should be kept in mind that the disease shows important heterogeneity and patient variability. As such, personalized treatment strategies may ultimately be essential for efficient management of the disease. Finally, sc sequencing combined with organoid technology (see below) could also lead to the conclusive identification of endometrium epithelial stem cells, and determine whether these cells are at the origin of endometriotic lesions.

### 4.3. Endometriosis Models

To uncover the pathogenic mechanisms underlying development and maintenance of endometriosis, several in vivo and in vitro research models have been (and are being) applied.

Because rodents are physiologically different from humans and do not show menstruation, artificial mouse models have been developed to study adhesion and growth of endometrial cells at ectopic sites. Therefore, endometrial tissue from mouse or humans is transplanted to ectopic sites in (immunodeficient) mice (mostly the peritoneal cavity), leading to endometriotic-like lesions [129]. However, lesions are mostly superficial and deep infiltrating lesions are not formed. Menstruating rodent models, mimicking human endometrium breakdown, have also been developed for endometriosis research [130,131]. In menstruating mouse models, hormone levels are manipulated via daily estrogen injections and implantation of a progesterone pellet [132]. Additional stimulation is achieved by administering sesame oil. ‘Menstruation’ then occurs when the progesterone pellet is removed. This menstrual tissue is implanted in syngeneic mice to induce endometriosis. However, this model still shows important limitations, such as variable endometrial response and the need for hormonal treatments, which appear different in different models, making comparisons and interpretation not straightforward. Interestingly, a mouse species has recently been discovered (i.e., the spiny mouse [133]) that also menstruates like humans, which will be an appealing research model to explore endometrium biology and diseases such as endometriosis.

Endometriosis can occur spontaneously in (menstruating) non-human primates such as the rhesus monkey and baboon, which, as a result, have been used to investigate mechanisms underlying pathogenesis and progression of the disease. For instance, in baboons, endometriosis is induced through intrapelvic injection of their own menstrual endometrium, leading to peritoneal lesions with similar histopathological features as human lesions [134,135]. However, use of non-human primates is ethically highly burdened.

In vitro, immortal(-ized) endometriotic cell lines (such as 12-Z) have been used to study pathogenetic pathways [136]. However, these cell lines are often carcinoma-derived, which are abnormal genome-wise and show aberrant hormone receptor expression and responses. Culturing primary epithelial and stromal cells from endometriotic lesions shows more physiological relevance, having revealed cellular and molecular differences between endometriotic, eutopic and healthy endometrial cells. However, these cells have only limited expansion potential while losing physiological characteristics, and the purity of each cell type’s culture is questionable. Both immortal and primary endometriotic cells have been used in co-culture settings, combining epithelial cells with stromal, peritoneal or immune cells. As another important limitation, the in vitro approaches lack the 3D configuration as present in vivo. Therefore, endometrial explants have been used to recreate the early stages of endometriosis development [137]; endometrial fragments were found to proliferate and invade into a fibrin-based matrix, leading to the generation of new glands, stroma and blood vessels, as occurs in endometriosis. However, use of explants suffers from high heterogeneity, undetermined cell composition and low throughput [138]. Spheroids have also been developed from endometriotic tissue, formed by the aggregation of dissociated cells, which were found to better mimic histological features of the lesion than 2D monolayer cultures [139]. However, developing spheroids needs substantial initial cell numbers, and the spheroids obtained are not highly expandable. Recently, organoids were established from patient endometriotic lesion biopsies, showing disease-associated traits including invasive activity and expression of MMPs and specific WNT pathway components as also found in the primary lesions [3]. Along this line, enriched gene expression was observed of pathways related to invasion, ECM remodeling, WNT signaling, as well as hormonal response, as compared to healthy endometrium-derived organoids. Moreover, endometriosis-derived organoids appear to recapitulate and retain the endometriosis-linked attenuated responsiveness to progesterone (such as the reduced expression response of the target genes 17β-hydroxysteroid dehydrogenase 2 (17HSDβ2), PAEP, SPP1 and LIF), as well as tissue-reproducing epigenetic changes (in particular, hypermethylation of the PR promoter leading to decreased expression of the longer isoform PR-B, and methylation changes in HOX cluster genes) [140,141]. The organoids, which strongly reproduce endometriotic lesions, now offer the ability to transcriptomically (e.g., using sc profiling), genetically (e.g., CRISPR/Cas gene editing) and functionally explore these and other pathways for their potential role in endometriosis, which may also lead to (new) therapeutic targets. The unlimited expansion potential of organoids, while retaining properties, and their cryopreserving and biobanking ability also renders this model a perfect fit for establishing drug screening platforms, even in a patient-personalized manner. Of note, it will also be important to enhance the model by combining the (organoid) epithelial cells with the stromal cells of the lesion, as well as with other cell types known to play a role, such as endothelial cells (angiogenesis) and immune cells (immune evasion; inflammatory nature), all proposed to play major roles in endometriosis pathology. Finally, microfluidic devices can also strongly help in modeling endometriosis-on-a-chip, to decipher epithelial–stromal crosstalk, angiogenesis and immune/inflammatory impact.

## 5. Conclusions and Future Perspectives

Achieving reliable models that closely mimic the endometrium is essential to uncovering the molecular and cellular mechanisms underlying its development and biological remodeling, at present not highly understood in humans. Within this context, the presence, identity and role of potential stem cells are far from clear. Organoids, typically developing out of tissue stem cells, are expected to be instrumental in solving this pressing question.

Mouse genetic models have enabled important knowledge to be gained regarding the molecular embryology of the uterus; however, so far, it is only translated to humans by looking into genetically screened patients with uterus maldevelopment. In vitro developmental models, starting from PSCs, are still in their infancy, while organoid designs appear more and more at the forefront to study the endometrium’s biology and pathology. Organoids provide a robust technology, recapitulating key features of the primary healthy or diseased tissue, highly useful for the investigation of biological and pathogenic mechanisms, as well as drug sensitivity and toxicity. Importantly, organoids have been shown to be amenable to modeling the endometrium menstrual cycle, and to recapitulate endometriotic lesions. Further advancement is expected from combining the organoid epithelial fraction with the other endometrial cell types in composite models to more closely understand the endometrium’s cellular and functional complexity (such as active crosstalk between cell types), as well as pathological issues (such as hormonal dysregulation and infertility in endometriosis). Such composite models will also achieve a clearer view on the role of the different cell types. With the rise of such 3D models in endometriosis research, the door to elucidating pathogenic mechanisms and investigating therapeutic advances is further opened.

## Figures and Tables

**Figure 1 jpm-12-01048-f001:**
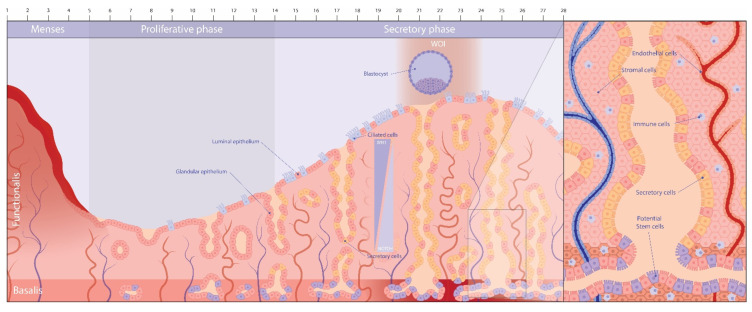
The endometrium during the menstrual cycle.

**Figure 2 jpm-12-01048-f002:**
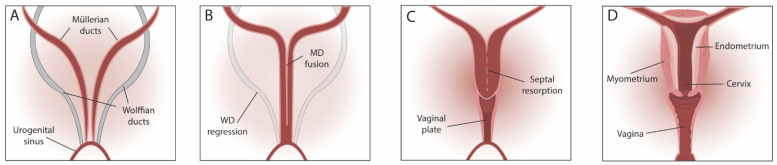
Embryonic development of the female reproductive tract. (**A**) Müllerian duct (MD) and Wolffian duct (WD) formation. (**B**) MD fusion and WD regression. (**C**) Septal resorption in the fused MDs. (**D**) Developed uterus with myometrium and endometrium.

**Figure 3 jpm-12-01048-f003:**
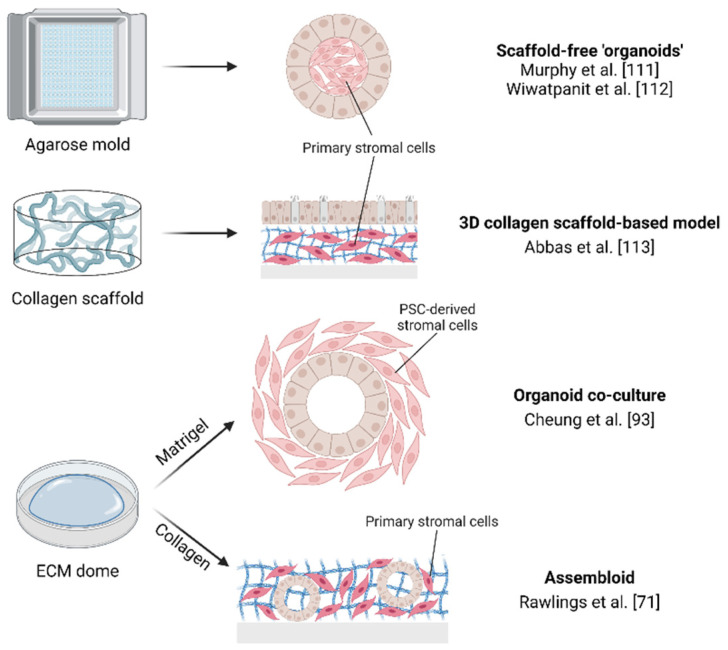
Endometrium composite models [1,2,3,4,5]. Created with BioRender.

**Table 1 jpm-12-01048-t001:** Factors involved in endometrium development.

Gene	Factor Encoded	Mutant Phenotype in	
		Mouse	Human
**Prenatal**			
*Formation of the MD*			
*Pax2*	Paired box TF	Absence of FRT [6]	
*Lim1* (*Lhx1*)	Homeodomain TF	Absence of FRT [7]	MRKH syndrome [13,14]
*Emx2*	Homeodomain TF	Absence of FRT [8]	
*Rarα*/*β*/*γ*	Retinoic acid receptors	Varying degrees of FRT defects [9]	
*Dlgh1*	Scaffolding protein	Aplasia of cervix and vagina [10]	
*Wnt4*	WNT secreted protein	Absence of FRT [11]	MRKH syndrome [15,16]
*Wnt9b*	WNT secreted protein	Absence of uterus and upper vagina [12]	MRKH syndrome [17]
*Differentiation of MD into FRT organs*			
*Wnt7a*	WNT secreted protein	Transformation of fallopian tube to uterus and uterus to vagina [18,19]	
*Hoxa10*	Homeodomain TF	Partial homeotic transformation of uterus to oviduct [20]	Defects in MD fusion [21,22]
*Hoxa11*	Homeodomain TF	Partial homeotic transformation of uterus to oviduct and hypoplastic uterus [23]	
*Hoxa13*	Homeodomain TF	Homeotic transformation of cervix to uterus; agenesis of caudal MD [24]	HFG syndrome [25]
*Tp63*	Tumor suppressor protein	Incorrect epithelial differentiation of lower genital tract [26]	SNPs associated with MD anomalies [27]
*TCF2 (HNF1ß)*	Homeodomain TF	NA	MRKH syndromeSevere genital malformations [28]
**Postnatal**			
*Wnt4*	WNT secreted protein	Reduction in endometrial glands [29]	
*Wnt5a*	WNT secreted protein	Absence of endometrial glands [30]	
*Wnt7a*	WNT secreted protein	Absence of endometrial glands [18]	
*Ctnnb1*	Signaling protein adhesion	Absence of endometrial glands [31,32]	
*Lef1*	TF	Absence of endometrial glands [33]	
*Timp1*	ECM modulator	Accelerated gland formation [34,35]	
*Igf1*	Growth factor	Hypoplastic uterus [36]	
*Foxa2*	Forkhead box TF	No gland differentiation [37]	
*Foxl2*	Forkhead box TF	Reduced stromal thickness, hypertrophic myometrium [38]	
*Dlx5/6*	Homeobox TF	Abnormal gland development [39]	
*Esr1*	Hormone receptor	Hypoplastic uterus [40,41]	

**Table 2 jpm-12-01048-t002:** Overview of single-cell profiling studies of the endometrium and endometriosis and derived candidate stem/progenitor cells.

References ^a^	Tissue Type	Candidate Stem/Progenitor Cell Markers	Clusters	
			Number	Cell Types
Endometrium				
Fitzgerald et al. [66]	Organoids	/	5 (13 subclusters)	Proliferative, epithelial, ciliated, unciliated and stem cells, and secretory cells after estrogen + progesterone exposure
Lucas et al. [67]	Primary	/	5	Epithelial, endothelial, immune and stromal (undifferentiated, decidual and senescent decidual) cells, and a distinct proliferative stromal subpopulation
Cochrane et al. [68]	Organoids	/	2	Ciliated and secretory cells
Wang et al. [69]	Primary	PDGFRB, MCAM, SUSD2 (Smooth muscle cell cluster contains cells expressing these MSC markers)	7	Stromal fibroblasts, endothelial, immune (macrophages and lymphocytes), ciliated and unciliated epithelial, and smooth muscle cells
Queckbörner et al. [70]	Primary	PDGFRβ, MCAM, SUSD2, (THY1)(However, these MSC markers were detected in all perivascular cells/pericytes)	7	Endothelial, epithelial, stromal, cycling stromal, two immune cell clusters (monocytes/macrophages and NK/T cells), and pericytes
Rawlings et al. [71]	Assembloids (epithelial + stromal cells)	/	11	Untreated: actively dividing and E2-responsive stromal cells; actively dividing, E2-responsive and ciliated epithelial cells. Hormonally treated: pre-decidual, emerging decidual and senescent decidual stromal cells; midluteal marker expressing and late-luteal marker expressing epithelial cells; transitional population
Garcia-Alonso et al. [72]	Primary	SOX9, LGR5	5 (14 subclusters)	Immune (lymphoid and myeloid), epithelial (SOX9^+^, luminal, glandular, and ciliated), stromal (decidualized and non-decidualized), endothelial (arterial and venous) and supporting cells (perivascular cells (PV STEAP4 and PV MYH11), smooth muscle cells and fibroblasts expressing C7
	Organoids	/	5 (10 subclusters)	Cell cluster from non-hormonally treated organoids (day 0, day 2, day 6, proliferative); estrogen-induced cells, ciliated (pre-ciliated and ciliated), secretory (secretory cycling, secretory), and KRT17^+^ cells
Lv H et al. [73]	Primary	/	15	Stromal, proliferating stromal, perivascular, luminal epithelium, glandular epithelium, ciliated epithelium, endothelial, mast, CD4+ T, CD8+ T, NKT, NK, peripheral blood-derived NK cells, lymphocytes, and macrophages
Endometriosis				
Ma et al. [74]	Primary (ectopic, eutopic and healthy)	/	9	Epithelial, endothelial, T, NK, mast cells, fibroblasts, macrophages/monocytes, neutrophils, and unknown
Fonseca et al. [75]	Primary (ectopic, eutopic)	FOXO1, XBP1, MAFF, JUND(regulons marking potential stromal stem/progenitor cells)	9 (96 subcluster)	Epithelial, endothelial, smooth muscle, myeloid, mast, B/plasma, T/NKT cells, fibroblasts, and erythrocytes
Tan et al. [76]	Primary (ectopic, eutopic, healthy) + organoids	MUC5B (epithelial cells; also express SOX9) SUSD2 (CCL19^+^ perivascular cells)	5 (58 subclusters)	Epithelial, stromal, endothelial, myeloid cells, and lymphocytes

^a^ Listed according to publication date.

## Data Availability

Not applicable.

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
