# Peer review of "Modeling Endometrium Biology and Disease"

_jpm, 2022, doi:10.3390/jpm12071048_

Round 1
Reviewer 1 Report
This review by Maenhoudt and colleagues explores the development and function of the endometrium, and the derivation of both in vivo and in vitro models to explore normal and pathogenic processes. The review is informative, well written and the figures are clear. I have only minor comments and suggestions to improve the work, including the inclusion of some key references and acknowledgment of recent fundamental changes in our understanding of the endometrium.
Line 15 – ‘start to shoot up’ I suggest a change of phrase here
Figure 1 gives the impression that during endometrial regeneration, the functionalis glands form exclusively by invagination of the lumen, which then grows downwards towards the basalis as the cycle progresses. However, whilst there may well be a contribution of luminal epithelium to functionalis gland regeneration in humans, it is widely hypothesised that the basalis glands give rise to the functionalis glands following menstruation. The figure should be edited to demonstrate a basalis origin of functionalis glands in the proliferative and early secretory phase. Furthermore, two recent studies (Tempest et al. 2020, Yamaguchi et al. 2021) have demonstrated that the basalis glands do not exist as a vertical continuation of the functionalis glands, but rather arrange horizontally along the endometrial-myometrial interface. The figure should be edited to reflect this major change in our understanding of endometrial architecture.
Line 146 – should be β-catenin
Line 239 – Reference 52 is from 2007; I suggest that more recent papers are also sited to capture the most up to date evidence for, and hypotheses on, basalis-resident endometrial stem cells
Line 255 – include references to support the notion of a stem cell niche in the basalis layer
The authors discuss the potential role of LGR5 positive stem/progenitor cells in mouse endometrium, but no evidence is presented for the role of LRG5 in human endometrium (Tempest et al. 2018)
Line 395 – PR-A and PR-B: define abbreviations
Line 428 – the original 1985 publication for Ishikawa cells would be useful here
Line 542 – whilst Turco et al. did not observe organoid formation from SSEA-1+ cells, spontaneous formation of gland-like spheroids (Valentijn et al. 2013) and immature gland structures (Hapangama et al. 2019) have been observed to form spontaneously from SSEA-1+ populations, which supports their progenitor phenotype
Section 3.2.3. The authors could add a few sentences on the complex histological architecture of the endometrium (basalis and functionals layers with differing gland configurations and stem/progenitor cell hierarchies) and how the ultimate ‘uterus in a dish’ would require the recreation of these elements to fully capture endometrial physiology in vitro.
Line 723 – should be TGFβ
Reviewer 2 Report
Endometrial stem cells studies and endometrial organoid modelling are at the forefront of unexplained infertility and endometriosis understanding. This article represents a real 'tour de force' and helps bring coherence in the field.
Author Response
We thank the reviewer for their positive feedback.
Reviewer 3 Report
The review by Maenhoudt et al. is well written and and focuses attention on a very current and very interesting topic.
I accept the manuscript in the present version
Author Response

(The authors gave the same response as above.)
